# An Analysis of Exhaust Emission of the Internal Combustion Engine Treated by the Non-Thermal Plasma

**DOI:** 10.3390/molecules25246041

**Published:** 2020-12-21

**Authors:** Ming-Hsien Hsueh, Chia-Nan Wang, Meng-Chang Hsieh, Chao-Jung Lai, Shi-Hao Wang, Chia-Hsin Hsieh, Tsung-Liang Wu, Jo-Hung Yu

**Affiliations:** 1Department of Industrial Engineering and Management, National Kaohsiung University of Science and Technology, Kaohsiung 807, Taiwan; cn.wang@nkust.edu.tw (C.-N.W.); shwang@nkust.edu.tw (S.-H.W.); charlie820906@gmail.com (C.-H.H.); wut@nkust.edu.tw (T.-L.W.); henry@nkust.edu.tw (J.-H.Y.); 2Institute of Undersea Technology, National Sun Yat-Sen University, Kaohsiung 804, Taiwan; pedro@g-mail.nsysu.edu.tw; 3Department of Fashion Design and Management, Tainan University of Technology, Tainan 71002, Taiwan; t30129@mail.tut.edu.tw

**Keywords:** non-thermal plasma (NTP), exhaust emission, internal combustion engine, ion chemical reaction

## Abstract

Industries’ air pollution causes serious challenges to modern society, among them exhaust gases from internal combustion engines, which are currently one of the main sources. This study proposes a non-thermal plasma (NTP) system for placement in the exhaust system of internal combustion engines to reduce the toxic contaminants (HC, CO, and NO_x_) of exhaust gases. This NTP system generates a high-voltage discharge that not only responds to the ion chemical reaction to eliminate NO_x_ and CO, but that also generates a combustion reaction at the local high temperature of plasma to reduce HC. The NTP system was designed on both the front and rear of the exhaust pipe to analyze the difference of different exhaust flow rates under the specified frequency. The results indicate that the NTP system can greatly reduce toxic contaminants. The NTP reactor placed in the front of exhaust pipe gave HC and CO removal efficiency of about 34.5% and 16.0%, respectively, while the NTP reactor placed in the rear of exhaust pipe gave NO_x_ removal efficiency of about 41.3%. In addition, the voltage and material directly affect the exhaust gases obviously. In conclusion, the proposed NTP system installed in the exhaust system can significantly reduce air pollutants. These results suggest that applying NTP to the combustion engine should be a useful tool to simultaneously reduce both emissions of NO_x_ and CO.

## 1. Introduction

The last few years have witnessed rapid development in industrial technology, with an increasing demand for fuel-based energy. The combustion of fuel is the primary source of energy for vehicle engines, aircraft engines, and industrial machinery. Fuel combustion produces a combination of chemical energy, thermal energy, and exhaust gases. Portions of combustion gases harmful to the environment such as unburned hydrocarbon, nitrogen oxides, carbon monoxide, and particulate matter are released into the environment. The exhaust gases emitted into the atmosphere cause air pollution due to the incomplete combustion of carbonaceous fuel. Inhaling gases are harmful to humans and cause health issues such as lung cancer, asthma, cardiovascular diseases [1,2,3]. Inhaling the exhaust gases damages the airway and may impair their function, especially the lungs [4]. Therefore, with mounting environmental concerns, the purification of exhaust gases from the internal combustion engine is becoming increasingly urgent to address. Balki et al. (2014) investigated the exhaust emission of an SI engine using gasoline, ethanol, and methanol at different engine speeds and observed that the use of alcohol instead of gasoline as the fuel in a low power engine caused a decline mainly in NO_x_, hydrocarbons (HC) and CO emissions. In addition, when methanol and ethanol were used in the combustion engine, the emissions of NO_x_, CO, and HC were reduced by 49% and 47.6%, 22.6% and 21.25%, and 21.6% and 19.13%, respectively [5]. Hsueh et al. (2016) studied a fuel temperature control device using a thermoelectric module (TEC) chip to measure engine performance and exhaust emission at various fuel temperatures and air/fuel (A/F) ratios. The emission ratios of HC and CO decreased but that of NO_x_ increased as the fuel temperature increased [6]. Doğan et al. (2017) investigated ethanol-gasoline blends as fuel for a four-cylinder and four-stroke spark-ignition engine and found that ethanol added to gasoline in the combustion engine fuel caused a reduction in CO and NO_x_ emission ratios [7].

In recent years, the development of non-thermal plasma (NTP) has attracted high interest and is considered as the green strategy for exhaust pollutant remediation. It can be used to remove various air pollutants such as SO_2_, NO_x_, HC, CO, and VOC [8,9]. In NTP, the electron temperature is higher than the gas temperature, which is close to room temperature. High energy electrons collide with surrounding gas molecules to produce reactive species such as dissociated molecules, ions, free radicals, and secondary electrons [10]. The reactive species actively combine with the environmental pollutants to form a decomposition product [11]. Various NTP systems such as the spark, dielectric barrier discharges, gliding arc, corona, microwave, and glow have been investigated for the direct conversion of exhaust gases [12,13,14,15,16]. The use of a plasma reactor with a catalyst to treat exhaust gases from gasoline engines was investigated by Dan et al. (2005), who observed that particulate matter removal efficiency ranged approximately from 25 to 57% [17]. Kim et al. (2017) investigated the effects of NTP on a lean premixed model gas turbine combustor of NO_x_ and CO by changing the mixing nozzle exit velocity and the equivalence ratio. The result showed a reduction in NO_x_ and CO emissions [18]. Adnan et al. (2017) tried to increase the rate of NTP decomposition by increasing the flow rates of the exhaust gases and could reduce the concentrations of CO, CO_2_, HC, and NO_x_ by more than 95% [19].

In this study, the variation of toxic contaminants in the exhaust system of the internal gasoline engine was investigated by using the NTP system. The NTP reactor was placed in the front or rear of the exhaust pipe to treat exhaust gas and the variation of emission was analyzed. The exhaust gas variation of the engine was observed by controlling the position, voltage, and types of NTP reactors with different engine speeds.

## 2. Experimental Apparatus and Techniques

### 2.1. Experimental Setup

The schematic of the experimental setup is shown in Figure 1, which mainly includes three sections, the production of exhaust gas by the motorcycle engine, the device for exhaust emissions reduction by the NTP system, and the exhaust gas analyzer of the detection system. The engine used for this experimental test was the model GP-125 manufactured by KYMCO Co. Ltd. (Taiwan), and there was no catalyst inside the exhaust pipe. The specifications of the engine in the power and torque measurement test are mentioned in Table 1. The experimental control conditions of the engine are mentioned in Table 2. The engine speeds were set in the range of 4000–6000 rpm, a frequently used range for practical motorcycle engines. The detailed experimental setup of the NTP system is shown in Figure 2, which is consisted mainly of three parts, the NTP reactor, the voltage converter, and the NTP control device. The NTP reactors used for this experiment test were model CR8EGP and CR8EIX manufactured by NGK Spark Plug Co., Ltd. (Aichi, Japan). The specifications of NTP reactors are mentioned in Table 3. NTP reactor was placed in the model A, in the front of the exhaust pipe or model B, in the rear of the exhaust pipe, as described in Figure 2 and Figure 3. The voltage converter used for this experiment test was model 3051A-LGL3–900 (KYMCO Co. Ltd., Taiwan). The NTP control device used for this experiment test was model DP-30032 (HILA International Inc, Taiwan). The specifications of the NTP control device are mentioned in Table 4. Pulsed voltages with a frequency of 10 Hz and adjustable amplitudes from 2 to 5 V were applied to the electrodes to produce a spark. The exhaust gas analyzer used for this experiment test was model EF-306EN (Exford, Taiwan), which measures the range of each exhaust gas, as mentioned in Table 5. The exhaust gas was measured at the end of the exhaust pipe and the data were recorded for 5 min under normal conditions to calculate an average. The A/F ratio was maintained at 13.7 to fix the stable fuel supply condition and observe the change in engine power performance and emission.

### 2.2. Experimental Procedure

The measurement equipment was adjusted and corrected before the experiment began. The preparation and measurement procedures were as follows:The engine condition of the motorcycle, such as engine oil, air filter, and fuel was checked.The exhaust gas analyzer was calibrated and warmed up.The engine was initiated for 30 min for the temperature to reach 100 °C.The NTP system was checked and the type of NTP reactor was selected.Before each experiment, a new NTP reactor was replaced.

In this study, the engine speed was 4000, 5000, and 6000 rpm, and the engine combustion produced exhaust gas when the engine started. The exhaust gas was discharged to the exhaust gas analyzer after passing through the NTP reactor. The data were directly recorded with the help of Microsoft Excel. The sparking frequency of the NTP control device was 10 Hz. The experimental procedure is presented in Figure 4. In these experiments, the different rpm of the engine were measured and compared to determine the optimal type and location of the NTP system. Based on the results of this experiment, the optimal condition of the NTP system was obtained and later applied to an actual vehicle.

The dissociation process of exhaust gas was defined as:

The removal efficiency of pollutants (RE) for HC, CO, and NO_x_:(1)$$\mathrm{RE}\;\left(\%\right)=\frac{\;{\left[\mathrm{Gas}\right]}_{\mathrm{off}}-{\left[\mathrm{Gas}\right]}_{\mathrm{on}}}{{\left[\mathrm{Gas}\right]}_{\mathrm{off}}}\times 100\%$$

The conversion efficiency of pollutants (CE) for CO_2_:(2)$$\mathrm{CE}\;\left(\%\right)=\frac{\;{\left[\mathrm{Gas}\right]}_{\mathrm{on}}-{\left[\mathrm{Gas}\right]}_{\mathrm{off}}}{{\left[\mathrm{Gas}\right]}_{\mathrm{off}}}\times 100\%$$
where [Gas]_off_ is the exhaust emission value without the NTP system; [Gas]_on_ is the exhaust emission value with the NTP system.

### 2.3. NTP Reactor

The spark is generated by the electrode gap of the NTP reactor. (Table 3 and Table 4). The combustible gas is ignited in the exhaust gas. The exhaust gas composition of engine combustion includes HC, CO, CO_2_, NO_x_, SO_2_, and PM2.5, etc. But the instrument used in this article can only measure HC, CO, CO_2_, and NO_x_. Therefore, the major reactions in the exhaust gas reduction by the NTP reactor are listed below [20,21,22,23].

The short high current spark with excited atomic radicals, excited molecules, and ions are generated, and the temperature during the spark phase can be as high as 3000 K. Therefore, we considered the Zeldovich thermal mechanism (see the equations below) of NO_x_ generation.
(3)$$N_{2}+O\to \mathrm{NO}+N$$
(4)$$N+O_{2}\to \mathrm{NO}+O$$

Oxygen molecules are broken down into oxygen radicals by the free electrons. The reactions of these oxygen molecules are listed below [20,21,22,23].
(5)$$O_{2}+e^{-}\to O+O$$
(6)$$O+O\to O_{2}$$
(7)$$O_{3}+e^{-}\to O_{2}+O$$
(8)$$O+O_{2}\to O_{3}$$
(9)$$O+O_{3}\to O_{2}+O_{2}$$

Under a higher energy electron collision, N_2_ transforms into N radicals and the water molecules partially dissociate to form active species, such as OH and H radicals. The results of H_2_O and N_2_ molecules at higher energy electron collision reactions are listed below [21,23].
(10)$$N_{2}+e^{-}\to N+N$$
(11)$$N+N\to N_{2}$$
(12)$$H_{2}O+e^{-}\to \mathrm{OH}+H$$
(13)$$H_{2}O+O\to \mathrm{OH}+\mathrm{OH}$$
(14)$$\mathrm{OH}+O\to O_{2}+H$$
(15)$$\mathrm{OH}+O_{3}\to {\mathrm{HO}}_{2}+O_{2}$$
(16)$$\mathrm{OH}+{\mathrm{HO}}_{2}\to O_{2}+H_{2}O$$

The mechanism for the destruction of CO_2_ in a non-thermal plasma involves the electron-induced dissociation of CO_2_ to give CO and O followed by recombination of the oxygen atoms to give the final products, CO and O_2_. CO can also recombine with atomic oxygen to reform CO_2_. The reactions are listed below [19,20].
(17)$${\mathrm{CO}}_{2}+e^{-}\to \mathrm{CO}+O$$
(18)$${\mathrm{CO}}_{2}+e^{-}\to C+O_{2}$$
(19)$$\mathrm{CO}+O\to {\mathrm{CO}}_{2}$$
(20)$$\mathrm{CO}+e^{-}\to C+O$$
(21)$$\mathrm{CO}+H_{2}O\to {\mathrm{CO}}_{2}+H_{2}$$
(22)$$C+O_{3}\to \mathrm{CO}+O_{2}$$

The process of NO_x_ removal mainly occurs through the oxidation pathway. Once N and O radicals are formed, the most important reactions of the oxidation of NO to NO_2_ are listed below [21,22,23].
(23)N+O→NO
(24)$$N+O_{3}\to \mathrm{NO}+O_{2}$$
(25)$$\mathrm{NO}+e^{-}\to N+O$$
(26)$${\mathrm{NO}}_{2}+e^{-}\to N+O+O$$
(27)$$\mathrm{NO}+O\to {\mathrm{NO}}_{2}$$
(28)$$\mathrm{NO}+O_{3}\to {\mathrm{NO}}_{2}+O_{2}$$
(29)$$\mathrm{NO}+N\to N_{2}+O$$
(30)$$\mathrm{NO}+H_{2}O\to {\mathrm{NO}}_{2}+2H$$
(31)$${\mathrm{NO}}_{2}+O_{3}\to {\mathrm{NO}}_{3}+O_{2}$$
(32)$${\mathrm{NO}}_{2}+{\mathrm{HO}}_{2}\to {\mathrm{HNO}}_{3}+O$$
(33)$${\mathrm{NO}}_{2}+\mathrm{OH}\to {\mathrm{HNO}}_{3}$$

## 3. Results and Discussion

### 3.1. HC

Figure 5, Figure 6, Figure 7 and Figure 8 present the HC emission value and the removal efficiency using different materials (Table 3) and at various locations (Figure 2 and Figure 3) of the NTP reactor. The HC emissions are known to be a result of the poor combustion of the fuel [24]. We found that the control voltage of 2 to 5 V in the NTP system could reduce the HC emissions at different engine speeds. The minimum HC removal efficiency was 1.2% at the control voltage 2 V for platinum (type one) at 6000 rpm and for reactor in the front of the exhaust system (model A). The control voltage 1 V exhibited little change in the HC removal efficiency, possibly because the voltage and the spark were too small. Therefore, it does not show in the figure. The control voltage 0 V indicates the original exhaust gas without the NTP system. Our results show that HC removal efficiency increased with increasing voltage because of the increase in shape and density of the spark with increasing voltage. With the increase in the contact area of the large spark and high density with the HC exhaust gas, the re-burning probability of the HC exhaust gas increased. Particularly, the shape and density of the spark approached the critical value at the control voltage 4 V, and the spark variation was small from the control voltage of 5 to 4 V.

Figure 9, Figure 10 and Figure 11 show the HC emission value and removal efficiency at control voltage 5 V using different materials and at various locations of the NTP reactor. Table 6 shows the maximum and minimum values of HC removal efficiency at control voltage 5 V using different materials and at various locations of the NTP reactor. The maximum HC removal efficiency was 34.5% for iridium alloy (type two) at 6000 rpm and for reactor in the rear of the exhaust system (model B). The minimum HC removal efficiency was 11.9% for platinum (type one) at 6000 rpm and for the reactor in the front of the exhaust system (model A). These results indicate that HC exhaust gas concentration increases with the increase in engine speed. This is because, with the increase in the engine speed, the gasoline fuel gets no time to burn in the engine. The NTP reactor materials of type one and type two reduce the HC exhaust gas concentration and type two exhibited a better effect than the type one. This is because the low resistance and small electrode area of type two material cause the formation of a larger spark with a higher internal density. With the increase in the contact area of large spark and high density with the HC exhaust gas, the re-burning probability of HC exhaust gas increased. The NTP reactor location in models A and B reduced the HC exhaust gas concentration. In model A, the HC removal efficiency increased with decreasing engine speed because of the slow flow rate of exhaust gas with decreasing engine speed. The re-burning probability of HC exhaust gas increased with the increase in contact time between the spark and HC exhaust gas. In model B, the HC removal efficiency increased with an increase in engine speed. This is because the flow rate of exhaust gas was almost the same in model B. The re-burning probability of HC exhaust gas increased with an increase in HC gas flow per unit area. The reduction in HC removal efficiency of model B was higher than that of model A. This is because of the larger cross-sectional area in model B, which leads to the complete release of spark and slow gas flow rate, increasing the re-burning probability of HC exhaust gas. Therefore, HC exhaust gas emissions get reduced.

### 3.2. CO

Figure 12, Figure 13, Figure 14 and Figure 15 present the CO emission value and removal efficiency of the different types and locations of NTP reactor. The control voltage of 2 to 5 V in the NTP system could reduce the CO emissions at different engine speeds. The minimum CO removal efficiency was 1.2% at the control voltage 2 V for Platinum (type one) and for in the front of the exhaust system (model A) at 6000 rpm. The results show that the removal efficiency of CO increases with increasing voltage. This is because with increasing voltage, the shape of the spark gets larger, leading to an increase in the production of free radicals. Abundant reactive radicals and higher combustion reaction temperature caused the increase in the dissociation probability of CO exhaust gas.

Figure 16, Figure 17 and Figure 18 present the CO emission value and removal efficiency of control voltage 5 V using different materials and at various locations of the NTP reactor. Table 7 shows the maximum and minimum values of CO removal efficiency at control voltage 5 V using different materials and at various locations of the NTP reactor. The maximum CO removal efficiency was 16.0% for iridium alloy (type two) at 5000 rpm and for reactor in the rear of the exhaust system (model B). The minimum CO removal efficiency was 3.3% for platinum (type one) at 6000 rpm and for reactor in the front of the exhaust system (model A). The NTP reactor material of types one and type two reduced the CO exhaust gas concentration. The effect of type two NTP reactor material was found to be better than that of type one because, low resistance forms larger spark and high internal density. The production of free radicals increases by increasing the spark size. The larger amount of radicals and higher combustion reaction temperature caused an increase in the dissociation probability of CO exhaust gas. The dissociation of CO is broadly divided into two parts. In the first part, the dissociation of CO gives carbon and oxygen atoms. Carbon atoms get deposited on the surface of the NTP reactor, as shown in Figure 19. The second part involves the recombination of CO and oxygen atoms to form the CO_2_. The location of the NTP reactor in models A and B reduce the CO exhaust gas concentration. In model A, the CO removal efficiency increased with a decrease in the engine speed, which caused the flow rate of exhaust gas to be slow. The dissociation probability of CO exhaust gas increased with the increase in contact time between the spark and CO exhaust gas. The CO removal efficiency was the highest in model B at 5000 rpm because the flow rate of exhaust gas was almost the same in model B. The dissociation probability of CO exhaust gas increased with an increase in CO gas flow per unit area. The reduction in CO removal efficiency of model B is higher than that of model A. This is because in model B, the dissociation of free radical and CO exhaust gas has a larger space, and the flow rate of exhaust gas was slow. This reduced the CO exhaust gas emissions. However, the amount of change in the CO exhaust gas measurement is small, there may be random errors in this experiment which may be caused by ambient temperature, humidity and dust. The random errors of CO exhaust values is about 0.1.

### 3.3. CO_2_

Figure 20, Figure 21, Figure 22 and Figure 23 present the CO_2_ emission value and conversion efficiency using the different materials and at various locations of the NTP reactor. The control voltage of 2 to 5 V in the NTP system could increase the CO_2_ emissions at different engine speeds. The maximum CO_2_ conversion efficiency was 0.9% at the control voltage 2 V for platinum (type one) at 4000 rpm and for reactor in the front of the exhaust system (model A). The results show that CO_2_ conversion efficiency increases with decreasing voltage. This is because decreasing voltage leads to a smaller shape of spark and this reduction causes a decline in the production of free radicals. Ultimately, this reduces the dissociation probability of CO_2_ exhaust gas.

Figure 24, Figure 25 and Figure 26 show the CO_2_ emission value and conversion efficiency at control voltage 5 V using different materials and at various locations of the NTP reactor. Table 8 shows the maximum and minimum values of CO_2_ conversion efficiency at control voltage 5 V using different materials and at various locations of the NTP reactor. The maximum CO_2_ conversion efficiency was 0.6% for platinum (type one) at 4000 rpm and for reactor in the front of the exhaust system (model A). The minimum CO_2_ conversion efficiency was −0.3% for iridium alloy (type two) at 4000 rpm and for reactor in the rear of the exhaust system (model B). Most of the NTP reactor materials of type one and type two increase CO_2_ exhaust gas concentration, while the type one exhibits a better effect than the type two. The reduced production of free radicals causes the dissociation probability of CO_2_ exhaust gas to decrease. CO_2_ is also obtained from the combination of CO and oxygen atoms. The dissociation of CO_2_ is divided into two parts. The first part involves the dissociation of CO_2_ to give CO and oxygen atoms. In the second part, the dissociation of CO_2_ gives carbon and oxygen atoms. Carbon atoms get deposited on the surface of the NTP reactor (Figure 19). Most of the NTP reactor locations of model A and model B increase the CO_2_ exhaust gas concentration. This is because the amount of CO_2_ dissociation is less than the amount of recombination of CO and oxygen atoms. This result is similar to that of CO exhaust gas emissions. However, the amount of change in the CO_2_ exhaust gas measurement was small, thus, there may be errors in measurement.

### 3.4. NO_x_

Figure 27 presents the oil temperature of the engine at different engine speeds. Figure 28, Figure 29, Figure 30 and Figure 31 present the NO_x_ emission value and the removal efficiency at different types and locations of the NTP reactor. The principal source of NO_x_ emissions is the oxidation of atmospheric nitrogen. We found that a control voltage between 2 to 5 V in the NTP system, could reduce NO_x_ emissions at different engine speeds. The maximum NO_x_ removal efficiency was 41.3% at the control voltage 4 V for iridium alloy (type two) at 3000 rpm and for reactor in the front of the exhaust system (model A). The minimum NO_x_ removal efficiency was 4.7% at the control voltage 2 V for platinum (type one) at 6000 rpm and for reactor in the rear of the exhaust system (model B). The results show that NO_x_ removal efficiency increased with increasing voltage, however, NO_x_ removal efficiency began decreasing after reaching control voltage 4 V. This is because the change in the spark of control voltage 5 V to the control voltage 4 V was small. The free radicals and exhaust gas degradation appeared to be similar, but the combustion reaction temperature increased. With the increase in combustion reaction temperature, the amount of diatomic nitrogen dissociated from the monatomic nitrogen increased and finally resulted in high NO_x_.

Figure 32, Figure 33 and Figure 34 present NO_x_ emission value and removal efficiency at control voltage 5 V using different materials and at various locations of the NTP reactor. Table 9 shows the maximum and minimum values of NO_x_ removal efficiency at control voltage 5 V using different materials and at various locations of the NTP reactor. The maximum NO_x_ removal efficiency was 37.6% for iridium alloy (type two) at 4000 rpm and for reactor in the front of the exhaust system (model A). The minimum NO_x_ removal efficiency was 7.5% for platinum (type one) at 6000 rpm and for reactor in the rear of the exhaust system (model B). The results show an increase in NO_x_ exhaust gas concentration with an increase in engine speed, which then causes the temperature of the combustion chamber to increase. At high temperatures, nitrogen and oxygen atoms easily recombine to form NO_x_. The NTP reactor materials of type one and type two reduce the NO_x_ exhaust gas concentration. The effect of type two material was found to be better than type one. The production of free radicals increased by increasing the spark size and the area of contact between the free radicals and NO_x_ exhaust gas also increased. This enhanced the probability of dissociation of NO_x_ exhaust gas to give nitrogen and oxygen atoms. The NTP reactor location in models A and B reduced the NO_x_ exhaust gas concentration. Moreover, the NO_x_ removal efficiency increased with decreasing engine speed because of the slower the flow rate of exhaust gas and the lower temperature of exhaust gas. The NO_x_ removal efficiency of model A was better among the two models because the products of NO_x_ dissociation can easily recombine to form the NO_x_ at high temperatures. The combination possibility of nitrogen and oxygen atoms is lower at the faster flow rate of the exhaust gas. Therefore, the NO_x_ exhaust gas emissions are reduced.

## 4. Conclusions

In this study, a motorcycle engine was examined using the NTP system to measure exhaust gas emission using different materials, voltage, and locations of the NTP reactor.

Considering the HC exhaust gas emissions, the NTP system re-burns HC exhaust gas to reduce the HC exhaust gas concentration. The HC removal efficiency increases with increasing voltage. The shapes and density of the spark approached critical values at the control voltage 4 V. The HC exhaust gas emissions of model B were lower than model A. The effect of type two NTP reactor material was better than that of type one. The maximum reduction of HC removal efficiency obtained with the NTP system reached 34.5%, while the minimum HC removal efficiency was 1.2%.

Considering the CO exhaust gas emissions, the reaction between free radicals and CO exhaust gas gates mainly dissociated to obtain carbon and oxygen atoms. Carbon gets deposited on the surface of the NTP reactor and affects the life of the NTP reactor. Furthermore, CO and oxygen atoms also recombine to form CO_2_. The CO removal efficiency increases with the increase in voltage. The CO removal efficiency of model B was found to be higher than that of model A. The effect of the type two NTP reactor material was better than type one. The maximum reduction of CO removal efficiency obtained with the NTP system reached 16.0%, while the minimum CO removal efficiency was 1.2%.

In terms of the CO_2_ exhaust gas emissions, the CO_2_ exhaust gas dissociates to form CO and oxygen atoms, and the dissociation of CO_2_ obtains carbon atoms and oxygen atoms. Like CO, carbon gets deposited on the surface of the NTP reactor. However, most of the amount of dissociation was found to be less than the amount of HC re-burning and CO reduction. Therefore, the total amount of CO_2_ increased and the CO_2_ conversion efficiency increased with decreasing voltage. The CO_2_ conversion efficiency of model A was higher than that of model B. The effect of type one NTP reactor material was better than that of type two. The maximum CO_2_ conversion efficiency obtained with the NTP system reached 0.9%, while the minimum CO_2_ conversion efficiency was −0.3%.

Considering the NO_x_ exhaust gas emissions, the NO_x_ gets dissociated by the NTP system to reduce the NO_x_ exhaust gas concentration. The removal efficiency NO_x_ increased with increasing voltage, but the NO_x_ exhaust gas removal efficiency began to decrease after reaching control voltage 4 V. As the combustion reaction temperature increased, the amount of diatomic nitrogen dissociated from the monatomic nitrogen increased and finally resulted in high NO_x_. The NO_x_ exhaust gas emissions of model A were lower than that of model B. The effect of type two NTP reactor material was better than that of type one. The maximum reduction of NO_x_ exhaust gas emissions removal efficiency obtained with the NTP system reached 41.3%, while the minimum NO_x_ removal efficiency was 4.7%.

Finally, according to the above discussion, the effect of iridium alloy is better than platinum for HC, CO, CO_2_, and NO_x_ emission removal. The effect of the NTP reactor in the rear of the exhaust system is better than in the front of the exhaust system for HC, CO, and CO_2_ emission removal. However, the NO_x_ emission removal for the reactor in front of the exhaust system is better than that in the rear of the exhaust system.

## 5. Patents

Taiwan Patent M526623.

## Figures and Tables

**Figure 1 molecules-25-06041-f001:**
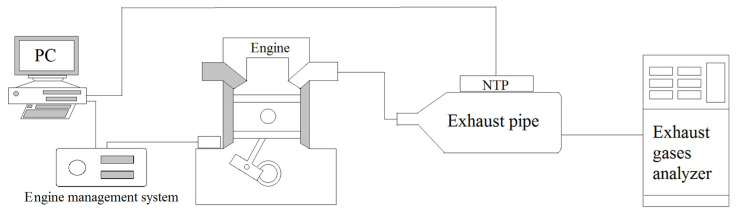
Schematic of this experiment and setup location.

**Figure 2 molecules-25-06041-f002:**
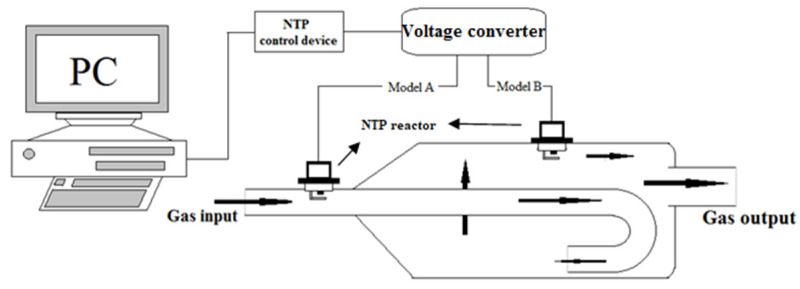
The setup of non-thermal plasma (NTP) system.

**Figure 3 molecules-25-06041-f003:**
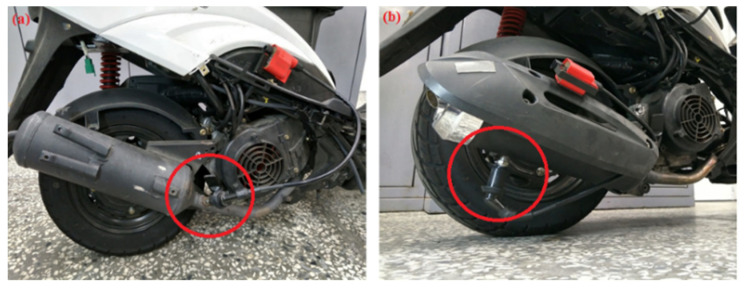
Photographic view of the device set up on (**a**) model A and (**b**) model B of the experimental motorcycle.

**Figure 4 molecules-25-06041-f004:**
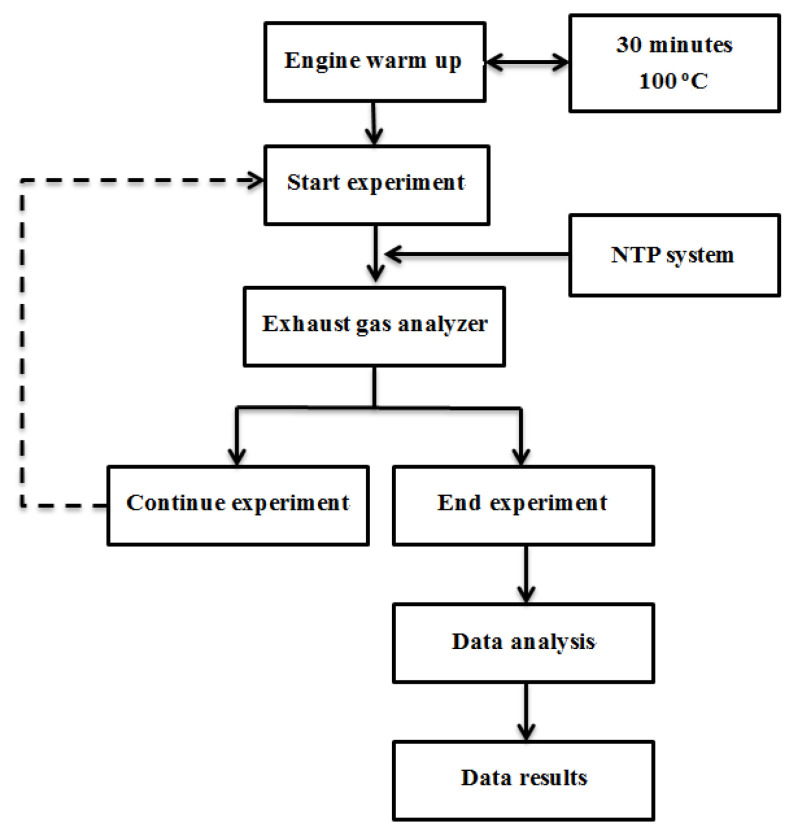
The procedure of experimental.

**Figure 5 molecules-25-06041-f005:**
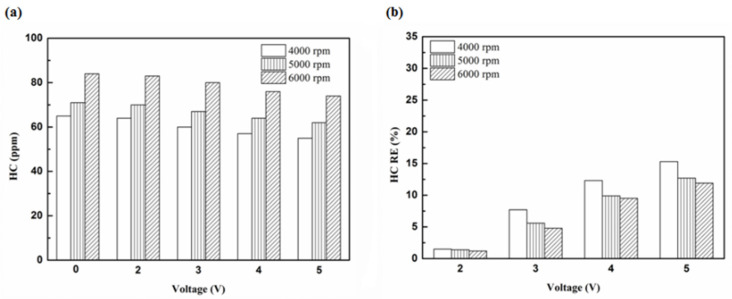
The HC (**a**) emission values and (**b**) removal efficiency (RE) using platinum in the front of the exhaust pipe.

**Figure 6 molecules-25-06041-f006:**
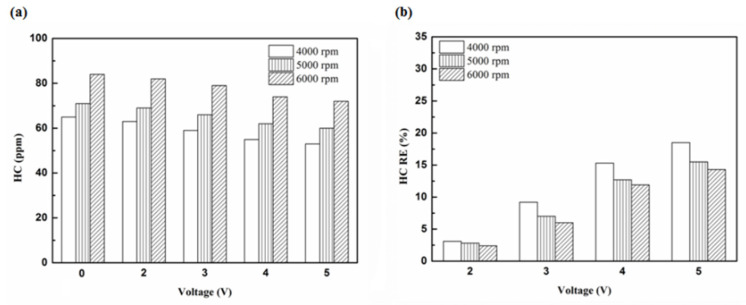
The HC (**a**) emission values and (**b**) RE using iridium alloy in the front of the exhaust pipe.

**Figure 7 molecules-25-06041-f007:**
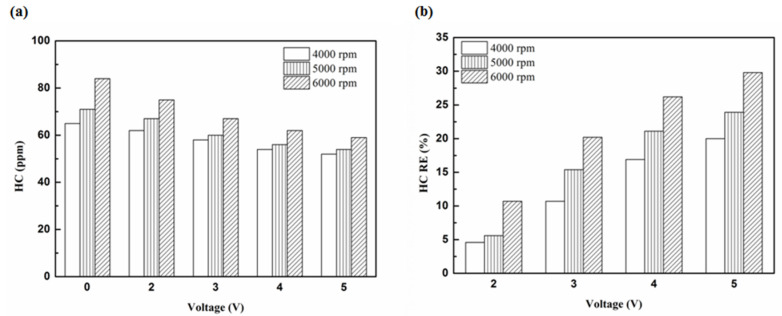
The HC (**a**) emission values and (**b**) RE using platinum in the rear of the exhaust pipe.

**Figure 8 molecules-25-06041-f008:**
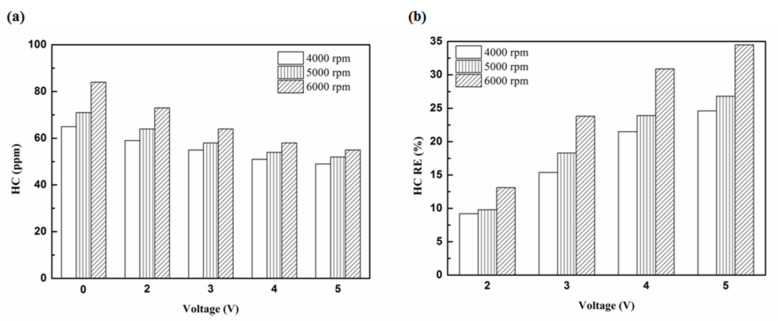
The HC (**a**) emission values and (**b**) RE using iridium alloy in the rear of the exhaust pipe.

**Figure 9 molecules-25-06041-f009:**
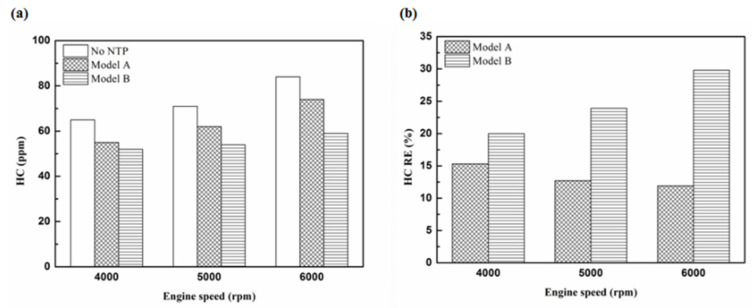
The HC (**a**) emission values and (**b**) RE of control voltage 5 V using platinum.

**Figure 10 molecules-25-06041-f010:**
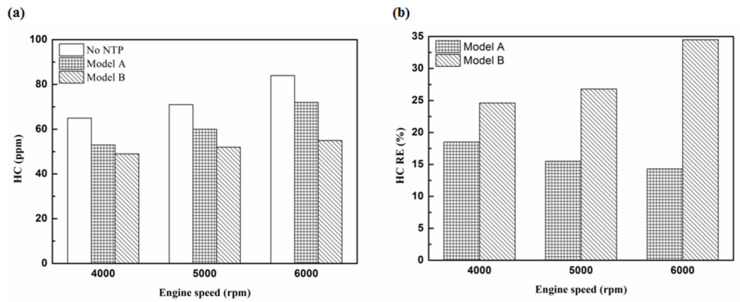
The HC (**a**) emission values and (**b**) RE of control voltage 5 V using iridium alloy.

**Figure 11 molecules-25-06041-f011:**
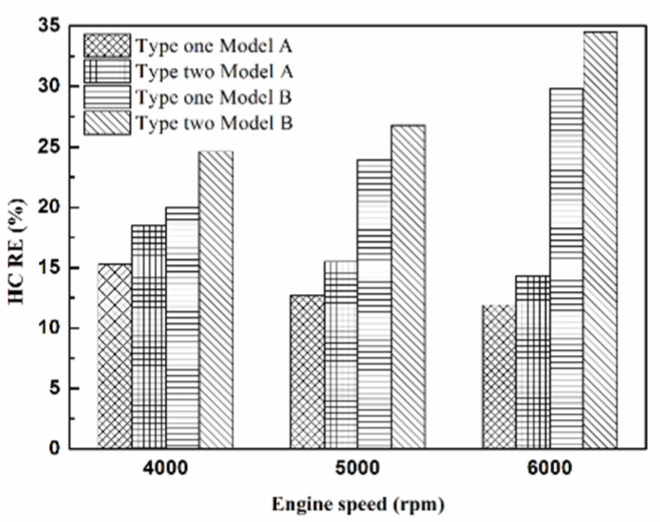
The HC RE at control voltage 5 V.

**Figure 12 molecules-25-06041-f012:**
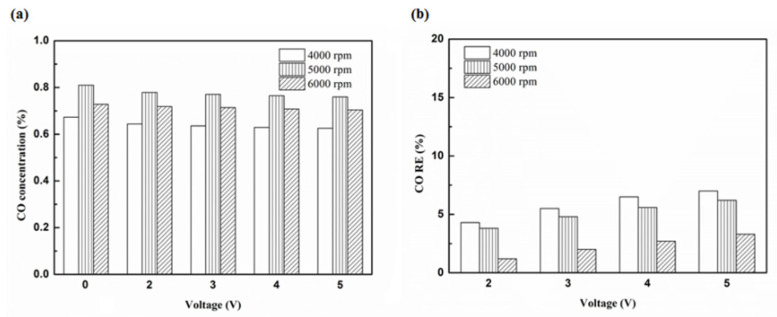
The CO (**a**) emission values and (**b**) RE using platinum in the front of the exhaust pipe.

**Figure 13 molecules-25-06041-f013:**
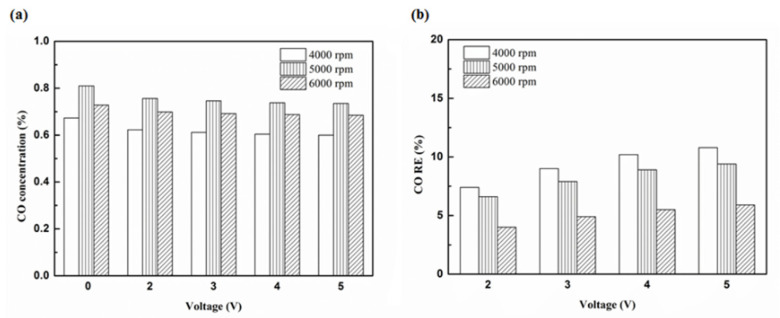
The CO (**a**) emission values and (**b**) RE using iridium alloy in the front of the exhaust pipe.

**Figure 14 molecules-25-06041-f014:**
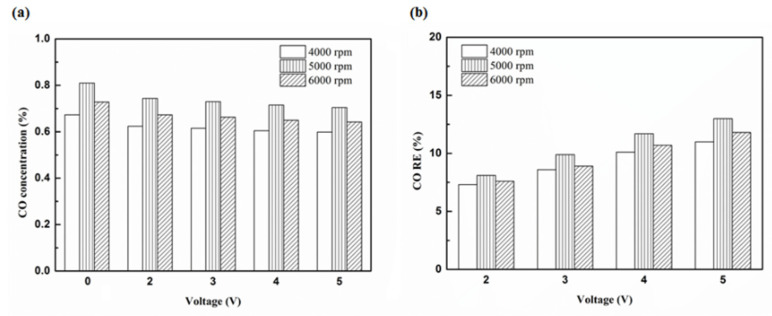
The CO (**a**) emission values and (**b**) RE using platinum in the rear of the exhaust pipe.

**Figure 15 molecules-25-06041-f015:**
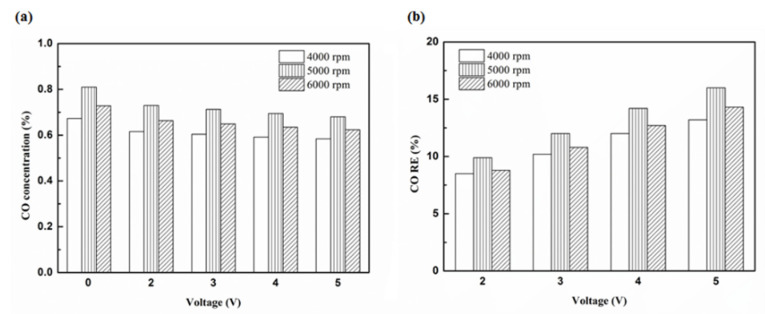
The CO (**a**) emission values and (**b**) RE using iridium alloy in the rear of the exhaust pipe.

**Figure 16 molecules-25-06041-f016:**
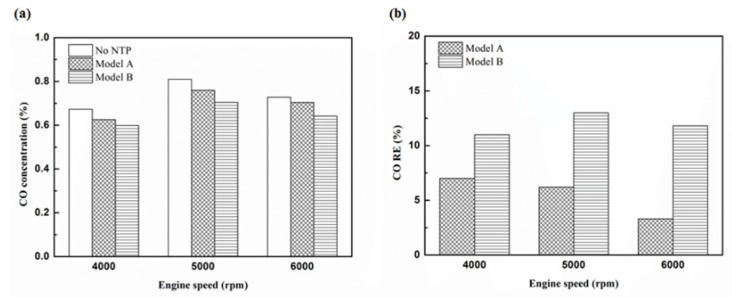
The CO (**a**) emission values and (**b**) RE of control voltage 5 V using platinum.

**Figure 17 molecules-25-06041-f017:**
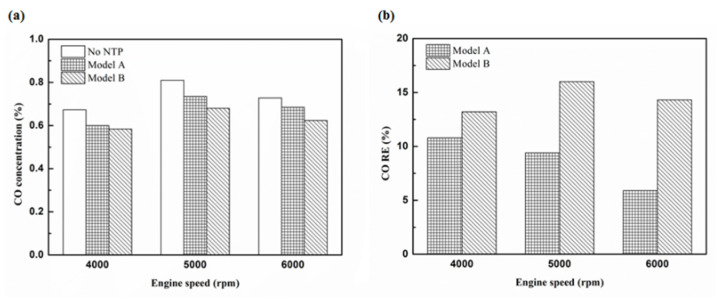
The CO (**a**) emission values and (**b**) RE of control voltage 5 V using iridium alloy.

**Figure 18 molecules-25-06041-f018:**
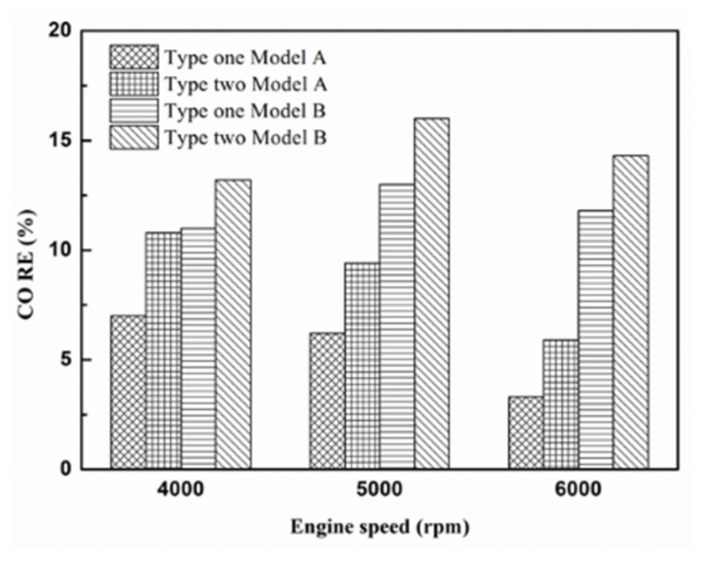
The CO RE at control voltage 5 V.

**Figure 19 molecules-25-06041-f019:**
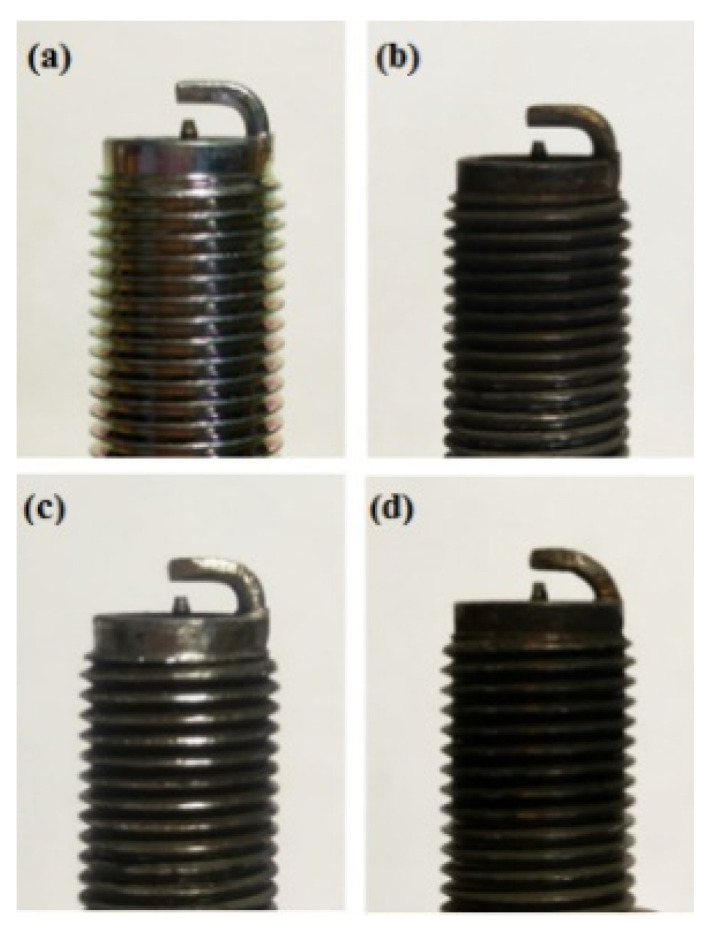
A photographic view of NTP reactor (**a**) platinum and (**c**) iridium alloy before this experiment and (**b**) platinum and (**d**) iridium alloy after this experiment.

**Figure 20 molecules-25-06041-f020:**
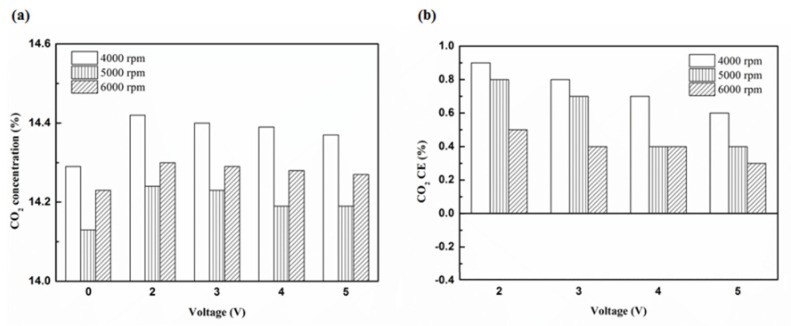
The CO_2_ (**a**) emission values and (**b**) CE at platinum in the front of the exhaust pipe.

**Figure 21 molecules-25-06041-f021:**
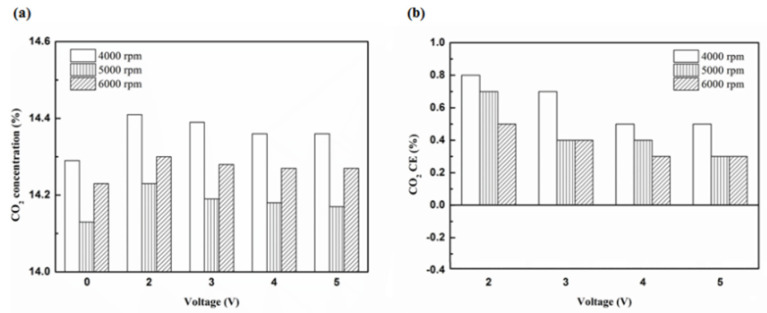
The CO_2_ (**a**) emission values and (**b**) CE using iridium alloy in the front of the exhaust pipe.

**Figure 22 molecules-25-06041-f022:**
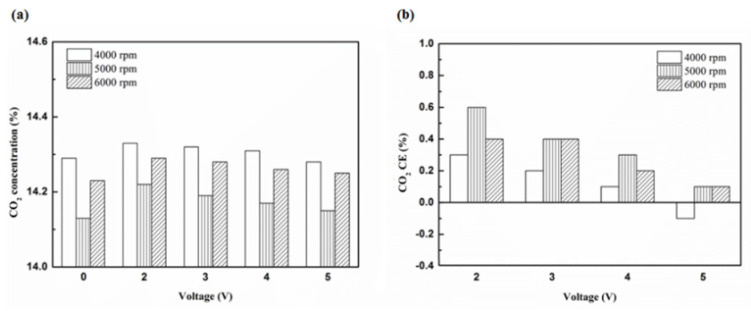
The CO_2_ (**a**) emission values and (**b**) CE using platinum in the rear of the exhaust pipe.

**Figure 23 molecules-25-06041-f023:**
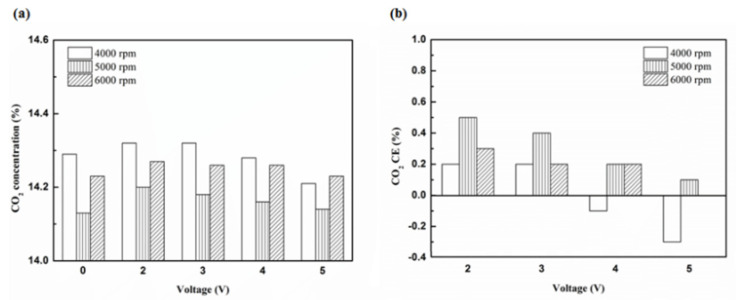
The CO_2_ (**a**) emission values and (**b**) CE using iridium alloy in the rear of the exhaust pipe.

**Figure 24 molecules-25-06041-f024:**
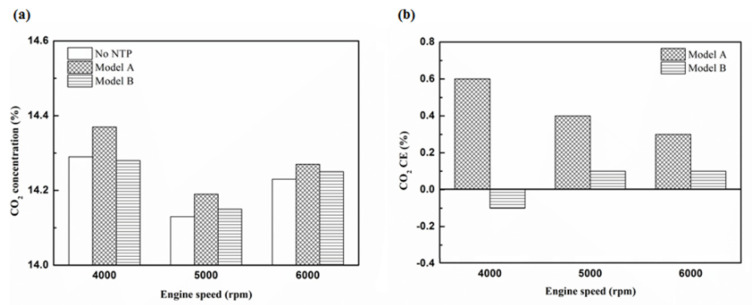
The CO_2_ (**a**) emission values and (**b**) CE of control voltage 5 V using platinum.

**Figure 25 molecules-25-06041-f025:**
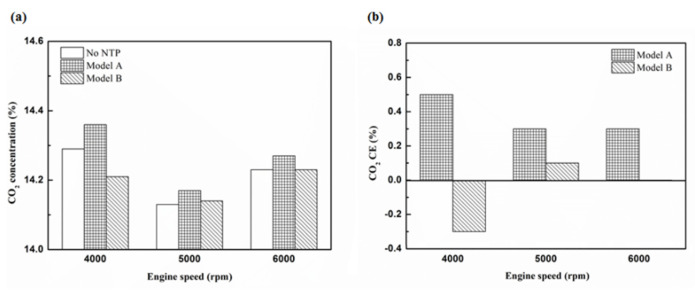
The CO_2_ (**a**) emission values and (**b**) CE of control voltage 5 V using iridium alloy.

**Figure 26 molecules-25-06041-f026:**
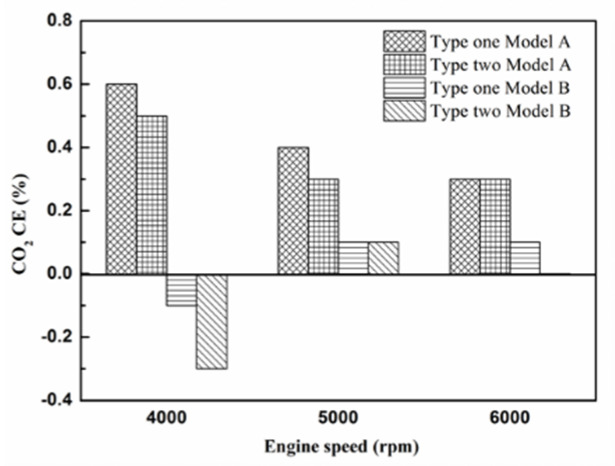
The CO_2_ CE at control voltage 5 V.

**Figure 27 molecules-25-06041-f027:**
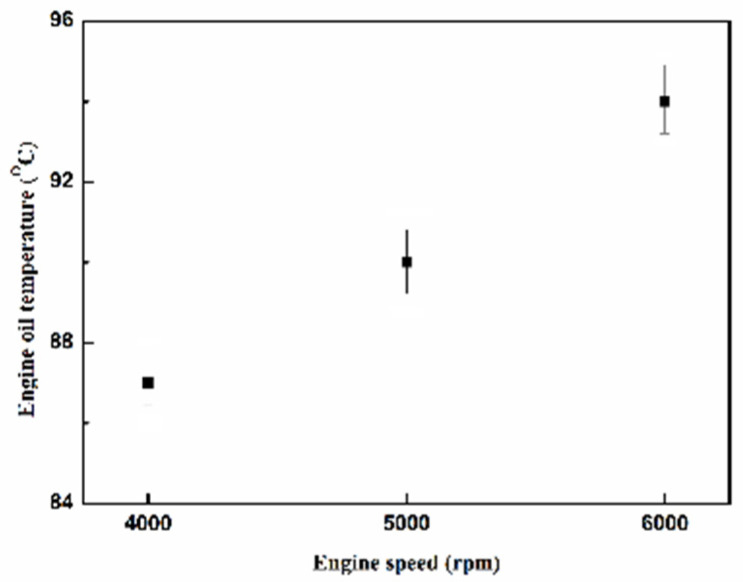
The oil temperature of the engine at different engine speeds.

**Figure 28 molecules-25-06041-f028:**
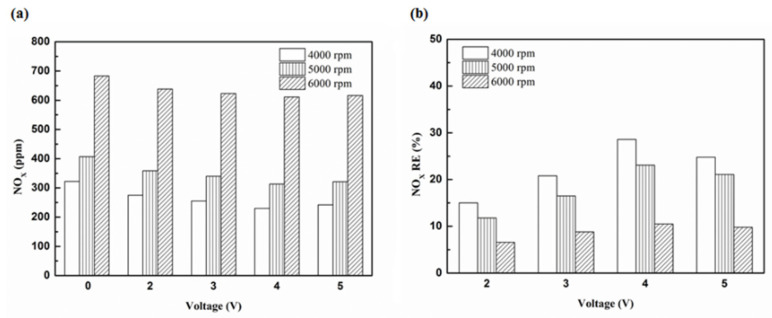
The NO_x_ (**a**) emission values and (**b**) RE using platinum in the front of the exhaust pipe.

**Figure 29 molecules-25-06041-f029:**
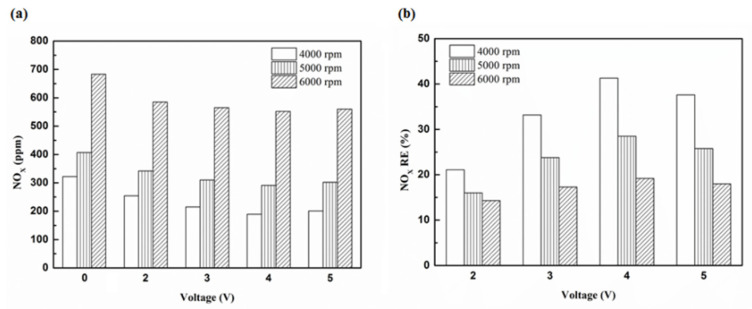
The NO_x_ (**a**) emission values and (**b**) RE using iridium alloy in the front of the exhaust pipe.

**Figure 30 molecules-25-06041-f030:**
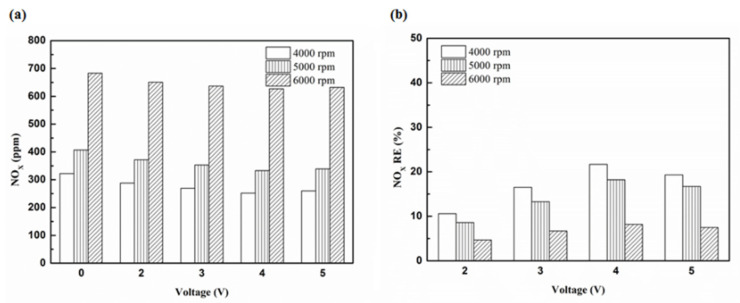
The NO_x_ (**a**) emission values and (**b**) RE using platinum and in the rear of the exhaust pipe.

**Figure 31 molecules-25-06041-f031:**
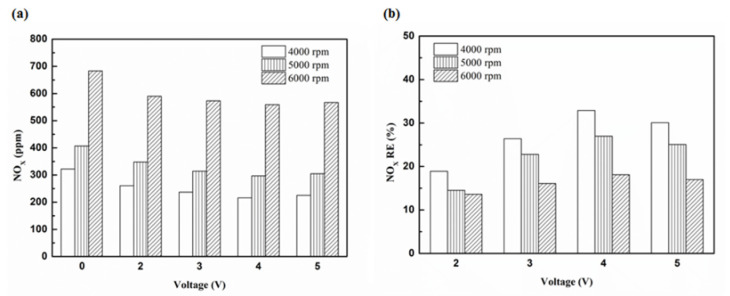
The NO_x_ (**a**) emission values and (**b**) RE using iridium alloy in the rear of the exhaust pipe.

**Figure 32 molecules-25-06041-f032:**
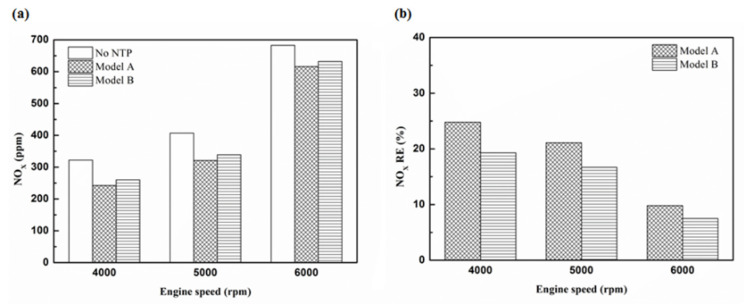
The NO_x_ (**a**) emission values and (**b**) RE of control voltage 5 V using platinum.

**Figure 33 molecules-25-06041-f033:**
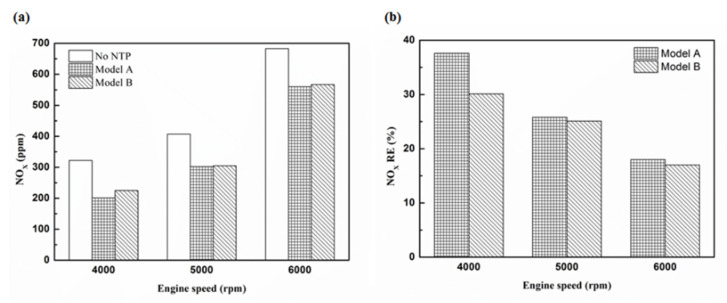
The NO_x_ (**a**) emission values and (**b**) RE of control voltage 5 V using iridium alloy.

**Figure 34 molecules-25-06041-f034:**
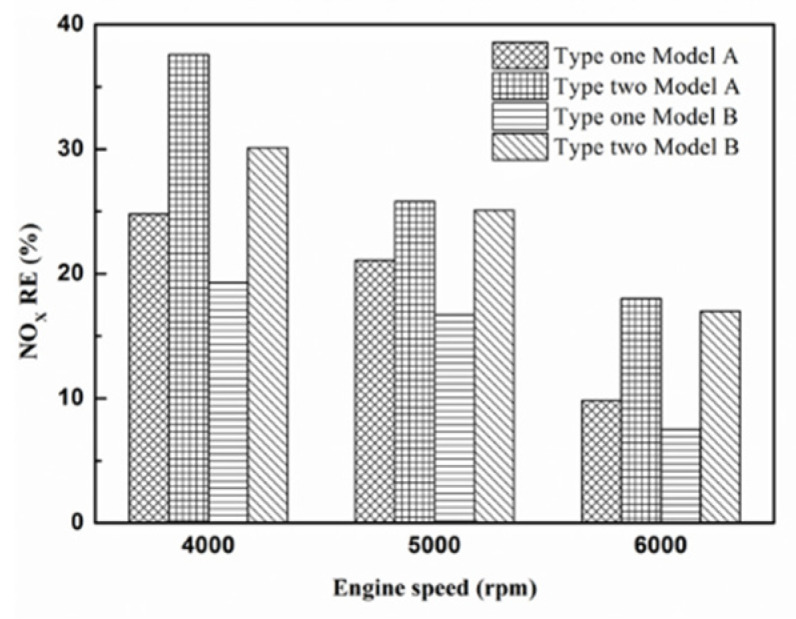
The NO_x_ RE at control voltage 5 V.

**Table 1 molecules-25-06041-t001:** The engine specification of motorcycle.

Parameter	Value
Model	GP125
Stoke	four-stoke
Engine type	Single cylinder
Displacement (c.c.)	124.6
Bore × Stoke (mm)	52.4 × 57.8
Compression ratio	9.9:1
Fuel	Unleaded gasoline
Max power (kw/rpm)	7.8/7500
Max torque (N-m/rpm)	10.3/5500
Wheelbase (mm)	1220
Air/fuel ratio (A/F)	13.7

**Table 2 molecules-25-06041-t002:** The control conditions of engine.

Parameter	Value
Engine speed (rpm)	4000, 5000, 6000
Fuel octane rating	95
Engine temperature (°C)	100–110
Intake air temperature (°C)	22–26

**Table 3 molecules-25-06041-t003:** The types of NTP reactor.

Parameter	Value	
	Type One	Type Two
Model	NGK-CR8EGP	NGK-CR8EIX
Material	Platinum	Iridium alloy
Discharging Gap (before used) (mm)	0.7	0.8
Discharging Gap (after used) (mm)	0.7	0.7
Center electrode diameter (mm)	0.8	0.8
Outer electrode diameter (mm)	2.20	1.85

**Table 4 molecules-25-06041-t004:** The specification of NTP control device.

Parameter	Value
Model	DP-30032
Main output voltage (V)	0–30
Main output current (A)	0–3
Fixed output voltage (V)	2.5/3.3/5
Fixed output current (A)	3
Resolution (mV/mA)	100/10
Precision	±(1% reading + 2 digits)
Power supply (V, Hz)	AC110/220 ± 10% selectable, 50/60
Dimensions (mm)	250W × 150H × 310D

**Table 5 molecules-25-06041-t005:** The range of each exhaust gas.

Parameter	Value	
	Measuring Range	Tolerance
HC (ppm)	0–2000	±12
CO (%)	0–10	±0.06
CO_2_ (%)	0–20	±0.5
NO_x_ (ppm)	0–5000	±25

**Table 6 molecules-25-06041-t006:** The maximum and minimum values of HC removal efficiency at control voltage 5 V.

Parameter	HC RE Value	
	Maximum	Minimum
Platinum and in the front of the exhaust pipe (Type one and model A)	15.3%	11.9%
Iridium alloy and in the front of the exhaust pipe (Type two and model A)	18.5%	14.3%
Platinum and in the rear of the exhaust pipe (Type one and model B)	29.8%	20%
Iridium alloy and in the rear of the exhaust pipe (Type two and model B)	34.5%	24.6%

**Table 7 molecules-25-06041-t007:** The maximum and minimum values of CO removal efficiency at control voltage 5 V.

Parameter	CO RE Value	
	Maximum	Minimum
Platinum and in the front of the exhaust pipe (Type one and model A)	7%	3.3%
Iridium alloy and in the front of the exhaust pipe (Type two and model A)	10.8%	5.9%
Platinum and in the rear of the exhaust pipe (Type one and model B)	13%	10.1%
Iridium alloy and in the rear of the exhaust pipe (Type two and model B)	16%	13.2%

**Table 8 molecules-25-06041-t008:** The maximum and minimum values of CO_2_ conversion efficiency at control voltage 5 V.

Parameter	CO_2_ CE Value	
	Maximum	Minimum
Platinum and in the front of the exhaust pipe (Type one and model A)	0.6%	0.3%
Iridium alloy and in the front of the exhaust pipe (Type two and model A)	0.5%	0.3%
Platinum and in the rear of the exhaust pipe (Type one and model B)	0.1%	−0.1%
Iridium alloy and in the rear of the exhaust pipe (Type two and model B)	0%	−0.3%

**Table 9 molecules-25-06041-t009:** The maximum and minimum values of NO_x_ removal efficiency at control voltage 5 V.

Parameter	NO_x_ RE Value	
	Maximum	Minimum
Platinum and in the front of the exhaust pipe (Type one and model A)	24.8%	9.8%
Iridium alloy and in the front of the exhaust pipe (Type two and model A)	37.6%	18%
Platinum and in the rear of the exhaust pipe (Type one and model B)	19.3%	7.5%
Iridium alloy and in the rear of the exhaust pipe (Type two and model B)	30.1%	17%

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
