# Peer review of "An Analysis of Exhaust Emission of the Internal Combustion Engine Treated by the Non-Thermal Plasma"

_molecules, 2020, doi:10.3390/molecules25246041_

Round 1

Reviewer 1 Report

This study seems to be a case study to reduce the air pollutants from fuel engine. The manuscript is well-wrote relatively. However, some kinds of misleadings and inconsistencies were indicated and explained. Author needs to make it clear.

  1.  1.What is TEC? (line 51)
  2. Some references were not cited (e.g. line 188 etc)
  3. Generally, if the RPM of the engine was increasing, torque also will be increasing. It means that fuel is comsumped more. Overal non-reactive combustion is relatevley higher than combustion fuel? Please make it clear.
  4. Also, CO results need to show the similar results with HC. However, it is different.
  5. CO2 generation is caused by combustion of fuel. Authors explained that the high RPM could could make a non-reactive combustion. However, the results did not show the huge different the CO2 emission. If the authors' claim is correct, CO2 emissions became to decreas, if the RPM will be increasing. (Fig. 20, 21, 22, 23 -(b))
  6. NOx is generated 3 mechanism as you know. However, it was just assumed to come from only fuel (it means just fuel NOx) when I read the indicated information and results. In this study needs to show the engine temperature each RPM condition.

Author Response

Dear reviewer:

Thank you very much for your letter dated 04 Dec 2020. We thank you for the time and effort that you have put into reviewing the previous version of the manuscript. your suggestions have enabled us to improve our work. Based on the instructions provided in your letter, we offer two versions. we uploaded the revised manuscript(Appendix 1)with all the changes highlighted in Adobe PDF.
Appended to this letter is our point-by-point response to the comments
raised by the reviewers. The comments are reproduced in the 1st column
and our responses are given directly in the 2nd column(Appendix 2).
We would like also to thank you for allowing us to resubmit a revised copy
of the manuscript.
We hope that the revised manuscript is accepted for publication.
Sincerely,

Reviewer 2 Report

In the pdf version of the article enclosed  I've introduced some remarks and proposition how to improve the paper. 

The title of the article should be shortened. In addition, the title is worded in such a way that it is not about analyzing the possibility of processing exhaust gases in the engine exhaust system, but rather about the use of non-thermal plasma to analyze exhaust gases in the engine exhaust system?, but the reviewer is not a native English speaker, and may not have right. I propose the following:

  1. An analysis of removing toxic contaminants from exhaust gases of the internal combustion engine by the non-thermal plasma,

or

  1. An analysis of exhaust emission of the internal combustion engine treated by the non-thermal plasma.

Line 19 - should be rather to reduce or to remove not to improve;

Tables 3 and 4 - When the reactor and NTP materials and locations are mentioned for the first time, table numbers should be quoted, and Figures 2 and 3, and then also when the test results are discussed.

Line 25 – after “…..34.5% and 16.0%”  should be respectively;

Line 130 and 132 - looking at the both equation (1) and (2) it is obvious that CE(%) + RE(%) = 0 or CE(%)= -RE(%), so what is the reason to define CE(%), would be simply stated that CE(%)= -RE(%)?

Line 138 - for article reader, it would be better to present the idea of the NTP reactor construction in more details;

Lines 144-184 - reviewer is not an expert in chemistry and chemical technology and was not able to confirm the correctness of the presented reactions. Moreover, the gas composition at the inlet of the exhaust system were not given;

Line 187 - after different materials - should be (Table 3) and after various locations of..... should be (Figure 2 and 3);

Line 268 - the measurement errors for each pollutant should be estimated;

The captions under almost all the drawings always refer the reader to type one and type two, as well as to model A and model B (they are shown at the Table 3 and Figure 2). 

In my opinion, it would be clearer to write directly below these drawings that it is platinum or iridium alloy as the electrode material and that the results are for the reactor at the front or rear of the exhaust system.

A lot of test results have been presented - for 4 pollutants (HC, CO, CO2 and NOx) - in 23 figures, but it is difficult to draw conclusions from them on which of the electrode material (Platinum or Iridium alloy) and what location of the reactor - in front or behind exhaust system, gives the best emission removal results for the tested internal combustion engine.

In conclusions, I propose to present a summary table, in which for the tested pollutants with both electrode materials and reactor locations, to provide the lowest and highest values of RE or CE.

Author Response

(The authors gave the same response as above.)

Round 2

Reviewer 1 Report

Revised manuscript is well modified. I think that it has enough to publish in Molecules.

This manuscript is a resubmission of an earlier submission. The following is a list of the peer review reports and author responses from that submission.